# Exogenous Sodium and Calcium Alleviate Drought Stress by Promoting the Succulence of *Suaeda salsa*

**DOI:** 10.3390/plants13050721

**Published:** 2024-03-04

**Authors:** Dong Zhang, Changyan Tian, Wenxuan Mai

**Affiliations:** 1State Key Laboratory of Desert and Oasis Ecology, Xinjiang Institute of Ecology and Geography, Chinese Academy of Sciences, Urumqi 830011, China; xjzhangdong@sina.cn; 2University of Chinese Academy of Sciences, Beijing 100049, China

**Keywords:** salt ions, leaf water content, leaf succulence, epidermal cells size, PEG, *Suaeda salsa*

## Abstract

Succulence is a key trait involved in the response of *Suaeda salsa* to salt stress. However, few studies have investigated the effects of the interaction between salt and drought stress on *S. salsa* growth and succulence. In this study, the morphology and physiology of *S. salsa* were examined under different salt ions (Na^+^, Ca^2+^, Mg^2+^, Cl^−^, and SO_4_^2−^) and simulated drought conditions using different polyethylene glycol concentrations (PEG; 0%, 5%, 10%, and 15%). The results demonstrate that Na^+^ and Ca^2+^ significantly increased leaf succulence by increasing leaf water content and enlarging epidermal cell size compared to Mg^2+^, Cl^−^, and SO_4_^2−^. Under drought (PEG) stress, with an increase in drought stress, the biomass, degree of leaf succulence, and water content of *S. salsa* decreased significantly in the non-salt treatment. However, with salt treatment, the results indicated that Na^+^ and Ca^2+^ could reduce water stress due to drought by stimulating the succulence of *S. salsa*. In addition, Na^+^ and Ca^2+^ promoted the activity of superoxide dismutase (SOD), catalase (CAT), and peroxidase (POD), which could reduce oxidative stress. In conclusion, Na^+^ and Ca^2+^ are the main factors promoting succulence and can effectively alleviate drought stress in *S. salsa*.

## 1. Introduction

Soil salinity is a global issue that is becoming increasingly severe, as salt negatively affects plants and reduces crop yields [1,2,3]. In crops, soil salinity causes hyperosmotic and hypertonic stresses that negatively affect various physiological processes, making all crops vulnerable to salinity, and even leading to their death [4,5,6]. The global salinized soil area continues to increase every year, except for naturally occurring soil salinity, which increases owing to irrigation practices and climate change, thereby expanding the area of salinized soils [7]. Euhalophytes are plants that have adapted to saline environments and have strategies to cope with salinity. In contrast, most crops are salt-sensitive glycophytes; therefore, improving them to generate salt-tolerant crops is a method to boost yield in saline agricultural areas [8,9,10].

Euhalophytes from the family Amaranthaceae can thrive under conditions in which the salt concentration is approximately 200 mM NaCl or higher, which would prove fatal to most conventional crops [11,12,13]. Growth in salinized soils can lead to water loss in plants owing to the decreased osmotic pressure in the soil. Euhalophytes respond to salt stress by accumulating osmolytes, which help to balance the cellular osmotic potential. Na^+^ and Cl^−^ are the primary components of the osmolyte NaCl [10,14], and accommodating high concentrations of Na^+^ and Cl^−^ in tissues is generally thought to be achieved by intracellular compartmentation [10]. Hence, numerous comprehensive reviews have been conducted on the salinity tolerance of halophytes in recent years. It is widely accepted that the ability of halophytes to efficiently sequester salt away from metabolically active tissues is the key trait that sets them apart from glycophytes [6,15,16,17,18,19,20]. Thus, the key traits of euhalophytes under salt stress need to be studied intensively.

Euhalophytes not only thrived in saline soil but also exhibited a noteworthy increase in their biomass when exposed to optimal salt content. Furthermore, the succulence of euhalophytes leaves (*Suaeda salsa*) has been observed to significantly increase with an increase in irrigation water salinity [21,22,23,24], which is linked to increases in cell dimensions, a reduction in surface area per tissue volume, and a significant amount of water content per unit of surface area [25]. This allows them to dilute toxic ion content in their cells, enabling them to withstand large quantities of salt [26]. Moreover, an investigation of the internal structure (mesophyll tissue and spongy parenchyma) of halophyte succulent organs further showed that the spongy parenchyma has a superior Na^+^ sequestration capacity compared to the mesophyll tissue and operates as a Na^+^ sink. This is achieved by a more efficient control of vacuolar Na^+^ retention resulting from the tighter regulation of slow vacuolar (SV) tonoplast channels, resulting in a better K^+^ retention capacity [20]. Therefore, studies indicate that succulence is a crucial trait for salt tolerance in euhalophytes [12]. Simultaneously, several reports have implicated genes that are involved in increasing plant tissue succulence which increases cell packing, decreases the intercellular air space within the palisade parenchyma, and increases the leaf water content in glycophytes (tobacco, grape, and Arabidopsis), thus diluting intercellular NaCl content within leaf tissues, resulting in improved salinity tolerance [27,28,29]. Engineered glycophyte tissue succulence might provide an effective strategy for improving salinity tolerance; therefore, a more detailed analysis of the formative factors of euhalophyte tissue succulence can provide a theoretical basis for the cultivation of salt-tolerant crops.

Different salts have different effects on the degree of succulence of euhalophytes. Compared to other salt treatments, NaCl leads to the maximum succulence degree [30,31]. In the investigation of *Salicornia europaea* and *S. salsa*, two types of euhalophytes, it was observed that under varying concentrations of NaCl, the degree of succulence in the organs (leaves and stems) of euhalophytes significantly increased in the presence of salt treatment. Furthermore, a positive correlation has been found between the degree of succulence and the size of epidermal cells [32]. Previous research has suggested that the accumulation of Na^+^ and Cl^−^ in the vacuole through the tonoplast Na^+^/H^+^ antiporter or vesicular transport can lead to a reduction in cell water potential [12]. This, in turn, facilitates water uptake in plants exposed to salinity, resulting in an increase in stem or leaf succulence [33]. Hence, the available evidence suggests that NaCl is the main factor promoting succulence in euhalophytes. However, soil-soluble salts in saline-alkali lands mainly contain inorganic ions, such as sodium and calcium, and the following associated anions: chloride, sulfate, and carbonate [34]. However, the effects of various salt ions on euhalophyte succulence have received limited attention [12]. Whether other salt ions participate in regulating euhalophyte succulence is still unknown, so further research is necessary to understand how various types and concentrations of salt affect euhalophyte succulence.

Plant tissue succulence is advantageous in salinity avoidance or mitigation and is associated with drought resistance [35]. Halophytism in succulents is the process of either diluting or sequestering salts at non-lethal concentrations in plant tissues [35]. The salts that accumulate are thought to be responsible for cell turgor, thereby promoting plant growth [36,37]. Xerophytism, on the other hand, is the plant’s ability to remain temporarily independent of the external water supply and is a common adaptation to drought [35,38,39]. Succulent organs can store large amounts of water, allowing them to maintain a high water potential and continue photosynthesis, even during extended periods of drought [40,41]. Thus, research showed that xerophytes, such as halophytes, counter drought stress through the succulence of their organs [36,39]. In addition, drought and soil salinity often occur together, both having a significant impact on the growth and productivity of plants [42,43], and halophytes are widely found in landscapes affected by soil salinity and water deficit [44]. Research has shown that early salt stress responses are due to general osmotic or drought stress and that sodium-specific responses are induced later [7]. However, previous research has revealed that euhalophytes (including *S. salsa*) cannot survive under dry conditions (induced by PEG6000) [45,46]. At the same time, research found that the biomass of euhalophytes significantly decreased when exposed to treatment with 40% field moisture capacity, in contrast to the 80% field moisture capacity treatment [32]. The interactions between different salt ions and drought conditions on euhalophyte succulence and growth have rarely been studied. Consequently, additional research is essential to elucidate the effects of the interaction between salt and drought stress on euhalophyte growth and succulence.

Two hydroponic culture experiments were conducted with *S. salsa*, an annual euhalophyte and a leaf-succulent halophyte, to gain a better understanding of the above. (ⅰ) Various sodium salts (NaCl and Na_2_SO_4_) and chlorine salts (MgCl_2_ and CaCl_2_) were used to determine the effect of distinct ions (Na^+^, Ca^2+^, Mg^2+^, SO_4_^2−^, and Cl^−^) on the succulence of *S. salsa* leaves treated with equal concentrations of chlorine and sodium ions. (ⅱ) Different concentrations of polyethylene glycol (PEG-6000) were used to simulate drought stress along with salt (NaCl, MgCl_2_, and CaCl_2_) treatments to determine the impact of drought stress and distinct ions (Na^+^, Ca^2+^, and Mg^2+^) on the succulence of *S. salsa* under treatment with equal concentrations of chlorine. This study provides a basis for knowledge on stress physiology and valuable information for engineering technologies to improve stress tolerance in important crop species.

## 2. Materials and Methods

### 2.1. Seed Collection

Seeds of *S. salsa* were collected from the halophyte botanic garden in Karamay City (45°28′6.38″ N, 84°59′41.61″ E), Xinjiang Province, China. The seeds were refrigerated at an approximate temperature of 4 °C before being utilised in experiments.

### 2.2. Plant Culture and Experimental Design

The experiment was conducted in a greenhouse located in Changji (44°09′59″ N, 87°04′56″ E), Xinjiang Province, China, with a 14 h light/10 h dark photoperiod (25 ± 3 °C days and 20 ± 3 °C nights). Black seeds were cultured with deionised water in seedling trays. After 15 days of germination, robust seedlings were selected and transferred to a 1 L black hydroponic box (127 × 87 × 114 mm) containing 1/2 Hoagland nutrient solution. The Hoagland nutrient solution contained 945 mg·L^−1^ Ca(NO_3_)_2_·4H_2_O, 506 mg·L^−1^ KNO_3_, 80 mg·L^−1^ NH_4_NO_3_, 136 mg·L^−1^ KH_2_PO_4_, 241 mg·L^−1^ MgSO_4_, 36.7 mg·L^−1^ EDTA-Fe, 0.83 mg·L^−1^ KI, 6.2 H_3_BO_3_, 16.9 mg·L^−1^ MnSO_4_·2H_2_O, 8.6 mg·L^−1^ ZnSO_4_·7H_2_O, and 0.025 mg·L^−1^ CuSO_4_·5H_2_O, and the pH value was 5.8 ± 0.2 (25 °C). The experimental treatments commenced after 15 days of plant establishment. Each treatment was replicated 8 times, and there were 2 seedlings in each hydroponic box.

### 2.3. Experiment 1: Different Salt Ions and Dosage Experiment

A two-factor randomised group experiment was conducted involving two distinct salt types: sodium salt (NaCl and Na_2_SO_4_) and chloride salt (MgCl_2_ and CaCl_2_). The concentrations of sodium and chloride ions were fixed at 0, 75, 150, and 300 mmol/L. A gradual increase in salinity, i.e., an increment of 1/5 of the final concentration every 2 days, was implemented in the hydroponic box to prevent osmotic shock. Once the desired salinity levels were reached, the treatments were maintained for an additional 30 days, with replacement of the nutrient solution every 2 days to ensure an adequate nutrient supply.

#### 2.3.1. Determination of Shoot Dry Weight, Leaf Water Content, and Succulence

Plant samples were rinsed with deionised water and separated into three components: roots, stems, and leaves. The fresh samples were weighed right away (FW) and placed in an oven at 105 °C for 30 min, and then baked at 60 °C for 72 h until a constant weight was achieved (DW). Leaf water content (LWC) was calculated as ((FW − DW)/DW) × 100%.

Ten mature leaves were collected, and their fresh weight (FW) was determined. The leaves were then oven-dried at 60 °C for 72 h, and their dry weight (DW) was measured. An Epson Expression 1600 Pro (Model EU-35, Epson, Tokyo, Japan) was used to transect the leaves at 300 DPI. Leaf surface area (LA) was calculated using the ImageJ 1.53T software (https://imagej.nih.gov/ij/) accessed on 25 November 2021. The succulence of the leaf (LS) was calculated as follows: LS (g H_2_O cm^−2^) = (FW − DW)/LA.

#### 2.3.2. Measurements of Epidermal Cells of Leaves

The leaf samples were collected and washed with distilled water to remove dust and salt. Subsequently, segments (5 mm long) were immersed in alcohol acetate formalin mixed fixative (FAA) solution, vacuum-infiltrated, and stored in a refrigerator at 4 °C. The leaves were then dehydrated using ascending concentrations of ethanol, critical-point-dried, and surface-coated with gold. The samples were placed on the stage of a Zeiss Supra 55 VP Scanning Electron Microscope (Carl Zeiss, Jena, Germany) for observation and photography. The measurements taken included the number of leaf epidermal cells per mm^2^ and their area (μm^2^). Samples were collected from three plants per treatment, and leaf tissues were collected from the same relative position on each plant. Approximately 20 locations were examined for each tissue sample replicate [32].

#### 2.3.3. Quantification of Na^+^, K^+^, Mg^2+^, Ca^2+^, Cl^−^, and SO_4_^2−^ Content

Harvested plant tissues were instantly cleaned with distilled water, dried in an oven at a temperature of 60 °C for 72 h, and then crushed into a fine powder using a mortar and pestle. The material was prepared by heating (~100 °C) 50 mg of plant material in 5 mL de-mineralised water for 2 h, The Na^+^, K^+^, Mg^2+^, Ca^2+^, Cl^−^, and SO_4_^2−^ concentrations were determined using ion chromatography (Dionex ICS2100, Thermo Fisher Scientific, Shanghai, China) [45].

#### 2.3.4. Determination of CO_2_ Assimilation Rate (A) and Stomatal Conductance (Gs)

Mature leaves at the same position were selected to measure the CO_2_ assimilation rate (A). A portable photosynthesis measurement system (LI 6400; LI-COR, Inc., Lincoln, NE, USA) was used to measure CO_2_ assimilation rate and stomatal conductance. Each treatment group was replicated six times.

### 2.4. Experiment 2: Drought-Salt Stress Experiment

According to the results of Experiment 1, a randomized group experiment with two factors was conducted, involving various salts (NaCl, MgCl_2_, and CaCl_2_) and four polyethylene glycol (NeoFroxx GmbH, Einhausen, Germany) treatments: 0%, 5%, 10%, and 15%. Different salts were added at an equal chloride ion concentration of 150 mmol/L. A control treatment (CK) that did not involve the addition of salt was included. A gradual increase in salinity and polyethylene glycol (PEG6000) in the hydroponic box, i.e., an increment of 1/3 of the final concentration every 2 days, was implemented to prevent osmotic shock. Once the desired salinity and polyethylene glycol (PEG6000) levels were reached, the treatments were maintained for an additional 10 days, with regular replacement of the nutrient solution every 2 days to ensure adequate nutrient supply.

Plant samples were rinsed with deionized water and separated into three components: roots, stems, and leaves. Subsequently, the shoot dry weight, leaf water content, and succulence were determined, and the leaf content of Na^+^, K^+^, Mg^2+^, Ca^2+^, and Cl^−^ was quantified. The freshly collected leaves were immediately stored in liquid nitrogen and then stored at −80 °C for measuring antioxidant enzyme activities and malondialdehyde content.

#### Determination of Antioxidant Enzyme Activities and Malondialdehyde Content

The contents of malondialdehyde (MDA) were determined by thiobarbituric acid. Enzyme-linked immunosorbent assay (ELISA) kits purchased from Suzhou Keming Biotechnology Co., Ltd. (Suzhou, China) were used to determine the superoxide dismutase (SOD; EC 1.15.1.1), catalase (CAT; EC 1.11.1.6), and peroxidase (POD; EC 1.11.1.7). SOD activity (1 U) was defined as the amount of enzyme required to reduce nitro blue tetrazolium chloride (NBT) to half of that in the control group. The activity of CAT (1 U) was expressed by the reduction of absorbance at 240 nm, while the activity of POD (1 U) was expressed by the reduction of absorbance at 470 nm.

### 2.5. Statistical Analysis

Two-way ANOVA was conducted using SAS 9.1™ software (SAS Institute, Inc., 1989, Cary, NC, USA) to examine the effects of salinity and concentrations on the biomass, leaf succulence, and leaf water content of S. salsa, as well as the effects of polyethylene glycol (PEG) concentrations and salinity on shoot dry weight, leaf succulence, leaf water content, ion concentrations, antioxidant enzyme activities, and malondialdehyde content. Principal component analysis (PCA), Pearson’s correlation coefficient, and linear analysis were performed using Origin pro 2023b (Origin Lab Corporation, Northampton, MA, USA), and the averages of three replicates were compared using the least significant difference (LSD) test, with a significance level of *p* < 0.05.

## 3. Results

### 3.1. Effect of Different Salt Ions and Dosage on the Shoot Dry Weight, Leaf Water Content, Succulence Degree, Ion Concentration, and Epidermal Cell Size and Number of S. salsa

#### 3.1.1. Shoot Dry Weight

The shoot dry weight of *S. salsa* was significantly higher when treated with 150 m M·L^−1^ of sodium salt (NaCl and Na_2_SO_4_) and chloride salt (MgCl_2_ and CaCl_2_) compared to the no-salt treatment (0 mmol/L), with increases of 63.81%, 38.07%, 34.28%, and 15.75%, respectively (*p* < 0.05, Figure 1A). However, when treated with 300 mM·L^−1^ of sodium salt (NaCl and Na_2_SO_4_) and chloride salt (MgCl_2_ and CaCl_2_), the shoot dry weight of *S. salsa* showed a decrease in comparison to the 150 mM·L^−1^ salt treatment (*p* < 0.05). In comparison to the other salt (Na_2_SO_4_, MgCl_2_, and CaCl_2_) treatments, the shoot dry weight of *S. salsa* was notably greater when treated with NaCl (*p* < 0.05). 

#### 3.1.2. Leaf Succulence and Water Content

As illustrated in Figure 1C, the succulence of *S. salsa* leaves was significantly greater when exposed to 150 mM·L^−1^ of sodium salt (NaCl and Na_2_SO_4_) and chloride salt (MgCl_2_ and CaCl_2_), compared to the 0, 75, and 300 mM·L^−1^ treatment. Among these, with the 150 mM·L^−1^ salt treatment, the NaCl and CaCl_2_ treatments producing the highest degree of succulence at 75.38% and 77.96%, respectively, compared to the control (0 mM·L^−1^) (*p* < 0.05). The water content of *S. salsa* leaves increased significantly when exposed to NaCl, Na_2_SO_4_, and CaCl_2_ salt treatments (*p* < 0.05), with no significant difference under the MgCl_2_ treatment compared to the no-salt treatment (Figure 1B). Therefore, the subsequent data analysis was conducted on the different salt treatments with concentrations of 0 and 150 mM·L^−1^.

#### 3.1.3. Leaf Ion Concentration

The analysis of Na^+^, K^+^, Ca^2+^, Mg^2+^, Cl^−^, and SO_4_^2−^ concentrations in the leaves of *S. salsa* under different sodium (NaCl and Na_2_SO_4_) and chloride salt (MgCl_2_ and CaCl_2_) treatments revealed that the sodium content of the leaves of *S. salsa* was significantly increased under the NaCl and Na_2_SO_4_ treatments. In contrast, the K^+^, Ca^2+^, and Mg^2+^ concentrations were significantly decreased by the sodium salt treatment (*p* < 0.05). Additionally, NaCl treatment significantly decreased the SO_4_^2−^ concentration in *S. salsa* leaves (*p* < 0.05). Moreover, the MgCl_2_ and CaCl_2_ treatments significantly reduced the K^+^ and SO_4_^2−^ concentrations in *S. salsa* leaves (*p* < 0.05; Figure 2).

#### 3.1.4. Leaf Epidermal Cell Size and Number

Electron microscopy was used to analyse the size and quantity of epidermal cells in the leaves of *S. salsa*, with no salt (CK) as a control (Figure 3). It was observed that the NaCl, Na_2_SO_4_, MgCl_2_, and CaCl_2_ salt treatments significantly increased the size of epidermal cells by 154.43%, 122.24%, 45.81%, and 150.27%, respectively (*p* < 0.05; Figure 3A). In contrast, the number of epidermal cells in the leaves decreased when exposed to these salt treatments, with decreases of 61.22%, 55.32%, 31.84%, and 60.54%, respectively (*p* < 0.05; Figure 3B).

### 3.2. Leaf CO_2_ Assimilation Rate, Stomatal Conductance, and Water Use Efficiency

As illustrated in Figure 4A, the CO_2_ assimilation rate of *S. salsa* was significantly higher when treated with 150 mM·L^−1^ of sodium salt (NaCl and Na_2_SO_4_) and chloride salt (MgCl_2_ and CaCl_2_) compared to the no-salt treatment (0 mM·L^−1^), with increases of 51.99%, 38.34%, 33.17%, and 27.25%, respectively (*p* < 0.05). However, compared to the CK and MgCl_2_ treatments, the NaCl, Na_2_SO_4_, and CaCl_2_ treatments caused a significant decrease in the stomatal conductance of *S. salsa* (*p* < 0.05, Figure 4B), leading to a significant increase in water use efficiency (*p* < 0.05, Figure 4C). 

### 3.3. Correlation Analysis

Pearson’s correlation coefficient was applied to analyse the association between the shoot dry weight (DW), leaf ion concentration (Na^+^, Ca^2+^, Mg^2+^, K^+^, Cl^−^_,_ and SO_4_^2−^), leaf succulence (LS), leaf water content (LWC), epidermal cell size (ES) and number (EN), and CO_2_ assimilation rate (A), stomatal conductance (Gs), and water use efficiency (WUE) of *S. salsa* under different sodium salt (NaCl and Na_2_SO_4_) and chloride salt (MgCl_2_ and CaCl_2_) treatments (Figure 5). The results demonstrated that Na^+^ concentration had a strong positive relationship with the shoot dry weight, leaf succulence, leaf water content, epidermal cell size, CO_2_ assimilation rate, and water use efficiency of *S. salsa* (*p* < 0.001) under various sodium salt treatments (Figure 5A), and was negatively correlated with the number of epidermal cells, stomatal conductance, and K^+^ concentration in leaves (*p* < 0.001). In contrast, Cl^−^ concentration was positively correlated with the shoot dry weight, leaf succulence, leaf water content, and CO_2_ assimilation rate of *S. salsa* (*p* < 0.05).

There was a statistically significant positive relationship between Cl^−^ concentration and shoot dry weight, leaf succulence, and epidermal cell size in *S. salsa* (*p* < 0.05) under various chloride treatments (Figure 5B). Additionally, Ca^2+^ concentration was found to have a significantly strong positive relationship with leaf succulence, leaf water content, epidermal cell size, and water use efficiency of *S. salsa* (*p* < 0.001), and was negatively correlated with the number of epidermal cells and stomatal conductance in the leaves (*p* < 0.001). However, no significant correlation was observed between Mg^2+^ concentration and the degree of succulence, water content, epidermal cell size, or quantity of *S. salsa* leaves.

### 3.4. Effects of Polyethylene Glycol (PEG) Concentration and Different Salt Ions on the Shoot Dry Weight, Leaf Water Content, Succulence Degree, Ion Concentration, Antioxidant Enzyme Activities, and Malondialdehyde Content of S. salsa

#### 3.4.1. Shoot Dry Weight

The biomass of *S. salsa* was significantly affected by the concentration of polyethylene glycol (PEG) concentration, salt ions, and their interaction (*p* < 0.001; Table 1). When PEG was used to simulate drought stress, the shoot dry weight of *S. salsa* decreased with increasing PEG concentration (*p* < 0.05). Without PEG (PEG0%), the biomass of *S. salsa* was significantly higher with the 150 mM·L^−1^ NaCl, MgCl_2_, and CaCl_2_ salt treatments than with no salt (CK), increasing by 58.52%, 46.78%, and 44.56%, respectively (*p* < 0.05). When different concentrations of PEG (5%, 10%, and 15%) were used to simulate drought stress, the biomass of *S. salsa* under the NaCl and CaCl_2_ salt treatments was higher than that under the no-salt (CK) and MC treatments (*p* < 0.05). Under the PEG15% treatment, there was no significant difference between the MgCl_2_ and CK treatments (Figure 6A).

#### 3.4.2. Leaf Succulence and Water Content

PEG concentration, salt type, and their interactions significantly affected the leaf water content and degree of succulence of *S. salsa* (*p* < 0.01; Table 1). When drought stress was simulated with PEG, the succulence of *S. salsa* leaves was significantly higher in the PEG0% treatment than in the PEG5%, PEG10%, and PEG15% treatments (*p* < 0.05; Figure 6C). Additionally, when different concentrations of PEG (0%, 5%, 10%, and 15%) were used in combination with NaCl and CaCl_2_, the degree of succulence of *S. salsa* leaves was significantly higher than that in the CK and MgCl_2_ treatments, with no significant difference between the CK and MgCl_2_ treatments (*p* < 0.05).

As drought stress (PEG) increased, the water content of *S. salsa* leaves under the CK and MgCl_2_ treatments decreased significantly (*p* < 0.05; Figure 6B); however, the water content of *S. salsa* leaves was not significantly different between the PEG5% and PEG10% treatments compared to the PEG0%, NaCl, and CaCl_2_ treatments (*p* < 0.05), and the NaCl treatment had a significantly greater water content than the MgCl_2_, CaCl_2_, and CK treatments (*p* < 0.05).

#### 3.4.3. Leaf Ion Concentration

The Na^+^ concentration in the leaves of *S. salsa* (Figure 7A) was significantly higher in the NaCl, MgCl_2_, and CaCl_2_ treatments than in the no-salt (CK) treatment. The Na^+^ concentration in *S. salsa* leaves treated with NaCl, MgCl_2_, and CaCl_2_ showed an initial increase and then a decrease with an increase in PEG concentration (*p* < 0.05), which increased 40.53, 45.78, 73.94, and 119.51 times compared to the CK treatment under different PEG concentrations (*p* < 0.05), whereas the Na^+^ concentration in the leaves of *S. salsa* treated with CK significantly decreased with an increase in PEG concentration.

The K^+^ concentrations in the leaves of *S. salsa* (*p* < 0.05; Figure 7B) decreased significantly with increasing PEG concentrations in all treatments. When treated with 0% and 5% PEG, the K^+^ concentrations in the leaves of *S. salsa* treated with salt (NaCl, MgCl_2_, and CaCl_2_) were significantly lower than those of the no-salt (CK) treatment, with no significant difference between NaCl, MgCl_2_, and CaCl_2_ treatments. When treated with 15% PEG, compared to the NaCl treatment, the CK and MgCl_2_ treatments significantly reduced the K^+^ concentrations in the leaves of *S. salsa* by 40.90% and 31.67%, respectively, with no significant difference between the NaCl and CaCl_2_ treatments (*p* < 0.05).

As the PEG concentration increased, the Ca^2+^ concentration in *S. salsa* leaves (Figure 7C) under the CK treatment noticeably decreased (*p* < 0.05). The NaCl, MgCl_2_, and CaCl_2_ treatments showed a pattern of initially increasing and then decreasing Ca^2+^ concentration (*p* < 0.05). At PEG15%, Ca^2+^ concentration in salt treatments (NaCl, MgCl_2_, and CaCl_2_) was significantly lower than that of PEG0%. Additionally, the Mg^2+^ concentration (Figure 7D) decreased with increasing PEG under different treatments (*p* < 0.05).

The Cl^−^ concentration in the leaves of *S. salsa* (*p* < 0.05; Figure 7E) significantly increased when exposed to increasing concentrations of PEG in the NaCl treatment. Compared to PEG0%, PEG5%, and PEG15%, the Cl^−^ concentration in the PEG10% treatment increased by 21.43%, 14.58%, and 4.82%, respectively (*p* < 0.05). In the CaCl_2_ treatment, the Cl^−^ concentration showed no significant difference between the PEG0%, PEG5%, and PEG10% treatments, but was significantly lower in the PEG15% treatment (*p* < 0.05). The MgCl_2_ treatment resulted in a significant decrease in the Cl^−^ content in the leaves of *S. salsa* when exposed to PEG10% and PEG15% (*p* < 0.05).

#### 3.4.4. Antioxidant Enzyme Activities and Malondialdehyde Content

Polyethylene glycol (PEG) concentration, salt ions, and their interactions significantly affected antioxidant enzyme activity and malondialdehyde content in *S. salsa* (*p* < 0.001; Table 1). As illustrated in Figure 8, the antioxidant enzyme activities (SOD, CAT, and POD) of *S. salsa* leaves were significantly higher in the NaCl and CaCl_2_ treatments than in the no-salt (CK) and MgCl_2_ treatments, leading to a decreased malondialdehyde content in the leaves (*p* < 0.05) under drought stress (PEG5%, PEG10%, and PEG15%).

### 3.5. Linear Analysis

Under drought stress conditions with different concentrations of PEG (0%, 5%, 10%, and 15%), a linear fitting analysis was conducted to examine the correlation between the degree of succulence of *S. salsa* leaves, water content, and aboveground biomass. The results revealed a significant positive correlation between the degree of succulence, water content, and shoot dry weight (*p* < 0.0001; Figure 9). This suggests that *S. salsa* could effectively reduce drought stress via leaf succulence.

## 4. Discussion

### 4.1. Na^+^ and Cl^−^ Promoted the Growth of S. salsa

For most euhalophytes, the ideal amount of salinity treatment promotes plant growth, and usually, a “curvilinear” growth response to external salinity is observed, with the highest growth rate occurring at intermediate salinities [19,32]. In this study, it was observed that the shoot dry weight of *S. salsa* increased and then decreased when exposed to different sodium salts (NaCl and Na_2_SO_4_) and chlorine salts (MgCl_2_ and CaCl_2_). Maximum biomass was seen at 150 mM·L^−1^ (Figure 1A). This result is consistent with the “curvilinear” growth trend [32], suggesting that adequate salt concentrations can facilitate the growth of halophytes and aid in completing their life cycle [47]. 

The biomass of *S. salsa* was the highest among the different salt treatments with NaCl (Figure 1A). Pearson’s correlation analysis revealed that Na^+^ and Cl^−^ were the primary driving forces behind the shoot dry weight of *S. salsa*, with Na^+^ having a greater effect than Cl^−^ (Figure 5). In addition, we found that the salt treatment (NaCl, Na_2_SO_4_, MgCl_2_, and CaCl_2_) significantly improved the carbon assimilation efficiency of the *S. salsa* under the condition of equal dosage (150 mM·L^−1^) of sodium and chloride ion (Figure 4A), and Na^+^ and Cl^−^ were the primary driving forces behind the carbon assimilation efficiency of *S. salsa*, which was unaffected by other ions (Ca^2+^, Mg^2+^, and SO_4_^2−^) (Figure 5). This result is consistent with previous studies showing that the photosystem of *S. salsa* can remain in its normal state in a saline environment, and the concentrations of certain ions that accumulate in the leaves can enhance the photosynthetic capacity, increasing the photoreaction and carbon assimilation efficiency [48]. This suggests that euhalophytes have adapted mechanisms that allow them to take advantage of high salt concentrations in their environment [49]. This may explain why Na^+^ and Cl^−^ were the main factors promoting *S. salsa* growth.

Euhalophytes are plants that can tolerate high levels of salinity and often display tissue succulence as a visible trait [12]. Research has found that the succulence of *S. salsa* leaves increases with increasing salt concentrations [24] and that there is a linear correlation between the succulence of euhalophytes (*Salicornia europaea*, *S. salsa*) and the dry mass of the shoot [32]. This suggests that it is possible to use the degree of tissue succulence as an indication of plant growth under salt stress. However, in our study, we found that there was no significant correlation between the succulence degree of *S. salsa* and shoot dry weight under different chloride salt (MgCl_2_ and CaCl_2_) treatments (Figure 5B). However, when different sodium salt (NaCl and Na_2_SO_4_) treatments were used, there was a significant positive correlation between the degree of succulence of *S. salsa* and shoot dry weight (Figure 5A). Field studies showed that the aboveground biomass of *S. salsa* had a “curvilinear” growth trend as the salt dosage increased, with the highest biomass at 20 g·L^−1^. In addition, the succulence of *S. salsa* leaves increased significantly with increasing salt dosage [24]. Thus, our results suggest that the degree of tissue succulence does not accurately reflect the growth state of halophytes under salt stress. This is because soil-soluble salts in saline-alkali lands are mainly composed of inorganic ions, such as sodium and calcium, and their associated anions, such as chloride, sulphate, and carbonate [34]. Therefore, it is assumed that different salt ions may have different impacts on the succulence of *S. salsa*, and thus further research is needed.

### 4.2. Na^+^ and Ca^2+^ Promoted the Degree of Succulence of S. salsa 

The capacity of succulent tissues to absorb large quantities of salt ions and store them in vacuoles, thereby decreasing or delaying the onset of salt stress, is a major factor in the survival of euhalophytes in salinized soils [12,26]. It has been suggested that NaCl is the primary contributor to the succulence of euhalophyte organs and that Na^+^ has a much greater effect than Cl^−^ [12]. In the present study, the succulence of *S. salsa* leaves substantially increased after exposure to different salts (NaCl, Na_2_SO_4_, CaCl_2_, and MgCl_2_) compared to the no-salt treatment (Figure 1C). The correlation analysis of the effects of different salt ions on the succulence of *S. salsa* leaves indicated that Na^+^, Ca^2+^, and Cl^−^ were the primary factors contributing to succulence development (Figure 5). Among these, Na^+^ and Ca^2+^ had a greater influence on the succulence of *S. salsa* leaves than Cl^−^. No correlation was found between the concentration of SO_4_^2−^ and Mg^2+^ and the succulence of *S. salsa* leaves (Figure 5). Therefore, in the present study, Ca^2+^ also contributed to the succulence of *S. salsa*.

Typically, succulent tissues (i.e., the tissues responsible for water storage) are caused by the growth of enlarged cells in either the photosynthetic tissue (chlorenchyma), a specialised achlorophyllous water storage tissue (hydrenchyma), or a combination of both [25,50,51]. In this study, electron microscopy was used to measure the size and number of epidermal cells in *S. salsa* leaves (Figure 3). The results showed that the area of epidermal cells in leaves treated with NaCl and CaCl_2_ was greater than that in the CK, Na_2_SO_4_, and MgCl_2_ treatments (Figure 3A); however, the number of epidermal cells decreased significantly (Figure 3B). Additionally, a strong positive correlation was observed between the epidermal cell size of *S. salsa* and the degree of succulence (Figure 5). Previous studies have suggested that the enlargement of epidermal cells is the primary factor in the formation of succulent organs in euhalophytes [32], further confirming that Na^+^ and Ca^2+^ are among the elements that increase the succulence of *S. salsa*.

Ma et al. argued that succulence could be caused by changes in cell size owing to improved osmotic adjustment [32]. In the present study, under the NaCl treatment, the concentrations of Na^+^ and Cl^−^ and the water content of *S. salsa* leaves were significantly higher than those in the no-salt treatments (Figure 2). It is widely accepted that in dicotyledonous halophytes, osmotic adjustment is the main factor in lowering the water potential, and this is mainly achieved by the accumulation of Na^+^ and Cl^−^ in the vacuole [11,14]. In addition, the CaCl_2_ treatment significantly increased the water content of the *S. salsa* leaves, as well as the concentration of Ca^2+^ and Cl, in this experiment (Figure 2). The findings of this study showed that the water content of the leaves had a positive relationship with Ca^2+^ and Na^+^, yet there was no significant association with other ions (Mg^2+^ and SO_4_^2−^) (Figure 5). Calcium, a key secondary messenger, is essential for signalling pathways and for regulating plant tolerance to both abiotic and biotic stresses [52,53]. Vacuoles are the primary storage sites for calcium in plants, and active Ca^2+^ transport systems, such as Ca^2+^ pumps and Ca^2+^/H^+^ exchangers, are responsible for moving cytosolic Ca^2+^ into the vacuole to maintain cytoplasmic calcium levels [54,55]. Thus, Ca^2+^ can be used as a regulator of inorganic osmosis in *S. salsa*.

Plant cells often contain high levels of intracellular osmolytes, and because of their rigid walls, they can become turgid without bursting [56]. A previous study suggested that the primary cause of succulent organs in halophytes is an alteration in the size of the organ epidermal cells [32]. Another study suggested that cell wall tension caused by turgor pressure is the reason for cell expansion [57]. In the present study, NaCl, Na_2_SO_4_, and CaCl_2_ treatments significantly decreased the stomatal conductance of *S. salsa* (Figure 4B), and Na^+^ and Ca^2+^ were negatively correlated with stomatal conductance (Figure 5). This is consistent with the results of previous studies, indicating that as salt dosage increases, the stomatal conductance of halophytes decreases significantly [14], and it is believed that this reduction in stomatal aperture is due to an increase in cytosolic Ca^2+^ [58]. 

Our results showed that adding NaCl and CaCl_2_ had a remarkable effect on the absorption of K^+^, leading to a reduction in the amount of K^+^ in the leaves (Figure 2). K^+^ is a critical factor in the control of stomatal aperture in the aerial parts of plants, as it is linked to a shift in osmolarity and the passage of ions, primarily K^+^, through the guard cell membrane [59]. The NaCl and CaCl_2_ treatments inhibit the absorption of potassium ions in *S. salsa*, which is a factor that decreases stomatal conductance. In addition, leaf succulence was negatively correlated with stomatal conductance and significantly positively correlated with leaf water content in *S. salsa* (Figure 5). Therefore, we concluded that the osmotic potential of halophytes was reduced by salt ions (Na^+^ and Ca^2+^), which led to continuous water absorption. Simultaneously, stomatal conductance is decreased by salt ions, creating an imbalance between water absorption and drainage and an increase in cell turgor pressure. Recent studies have shown that biological processes may actively regulate turgor pressure, indicating a more active role of turgor pressure in cell growth regulation. [56,60,61,62]. Therefore, we speculated that changes in cell size caused by cell turgor pressure may be responsible for succulence. 

The cell walls of succulent tissues are usually very thin and flexible [35,63], and their ability to swell is affected by the structure of the cell wall. The production of the load-bearing component of the cell wall, cellulose, and its crosslinking by pectin are modified when exposed to external salt [64]. Studies have revealed that the biomechanics and biochemistry of succulent leaves differ from those of non-succulent species because succulent species have highly elastic cell walls [65]. Furthermore, a glycomic analysis of succulent tissues has revealed several similarities across phylogenetically diverse succulent species, such as a higher degree of HG methyl esterification and a greater abundance of RG-I. These biochemical differences likely contribute to the high elasticity of the cell walls of succulent organs, which in turn facilitates cell deformation during salt and drought stress [37,66]. Some glycomic features appear exclusive to certain succulent lineages, indicating glycomic diversity among succulent plants [37].

### 4.3. Na^+^ and Ca^2+^ Can Alleviate Drought Stress of S. salsa

Drought stress affects plant growth by limiting water and nutrient uptake. In contrast, salt stress is caused by an abundance of neutral salts [67] that can impede growth owing to both osmotic stress and ionic toxicity [68]. Halophytes and xerophytes use strategies, such as organ succulence and osmotic adjustment, to counteract osmotic stress [35]. It is possible that the succulence of halophyte tissues is promoted by water (drought) stress. This study suggests that the succulence and water content of *S. salsa* leaves in the no-salt treatment (CK) decreased significantly as the degree of drought (PEG) increased (Figure 6C), indicating that water (drought) stress was not a factor contributing to the succulence of *S. salsa* leaves. A previous study reported that drought stress (induced by PEG6000) had a more serious negative effect on *S. salsa* growth [45,46]. We also found that the biomass of *S. salsa* drastically decreased when exposed to drought stress, suggesting that it is not drought-tolerant (Figure 6A).

Succulence is the capacity to facilitate water homeostasis and to buffer plants from the vagaries of a spatiotemporally unpredictable external water supply [69]. The linear analysis in this study showed a significant correlation between the biomass of *S. salsa* and its succulence when exposed to drought stress (Figure 9B), demonstrating that succulence is an effective strategy for overcoming drought stress in *S. salsa*. Experiment 1 showed that Na^+^ and Ca^2+^ were the main factors promoting the succulent organs of *S. salsa*. This study demonstrated that when chlorine concentrations (150 mM·L^−1^) were equal, the NaCl and CaCl_2_ treatments could maintain the water content (Figure 6B) in the leaves during mild (PEG5%) and moderate (PEG10%) drought stress by stimulating the formation of succulent organs (Figure 6C). However, the MgCl_2_ treatment could preserve leaf water content when exposed to mild drought (PEG5%). However, as the drought severity increased, the water content and biomass of the leaves significantly decreased, and leaf wilting occurred (Appendix A). Therefore, it can be deduced that Na^+^ and Ca^2+^ effectively alleviate drought stress in *S. salsa*.

The injurious effects of drought stress are mainly mediated via water potential stress mechanisms [67]. When confronted with osmotic stress, halophytes typically absorb large quantities of salt ions and sequester them in vacuoles to reduce the osmotic potential [70]. This study revealed that as the severity of drought increased, the concentrations of Na^+^ and Cl^−^ in the leaves of *S. salsa* under the NaCl treatment increased significantly, whereas the concentrations of Ca^2+^ and Cl^−^ in the CaCl_2_ treatment also increased significantly (Figure 7). Moreover, Na^+^ and Ca^2+^ significantly increased the water use efficiency (WUE) of *S. salsa* leaves (Figure 3C). It is believed that halophytes respond to drought stress by first stimulating the succulence of *S. salsa* organs through salt ions and then absorbing high levels of salt ions to reduce the osmotic potential, thus alleviating drought stress. 

Drought can cause an imbalance in reactive oxygen species levels and impair the antioxidant defence mechanisms of plants [71]. This can cause oxidative damage [72]. MDA is a biomarker of membrane lipid peroxidation in plants [73] and is usually observed in higher amounts under drought stress [74]. Our study revealed that the MDA content in the CK and MgCl_2_ treatments was notably higher than that in the NaCl and CaCl_2_ treatments when the plants were exposed to drought stress (Figure 8D). This implies that NaCl and CaCl_2_ can enhance the antioxidant capacity of *S. salsa* under drought stress. Antioxidant enzymes are involved in the response to drought or salt stress, and their activity reflects the ability of plants to tolerate various types of environmental stresses [75]. Halophytes have high antioxidant capacity, mainly because of their high antioxidant oxidase activity compared to that of glycophytes under salt stress [15]. Calcium (Ca^2+^), as a multifunctional messenger, reduces lipid peroxidation and increases the activity of antioxidant enzymes in response to drought stress [75,76]. This experiment revealed that NaCl and CaCl_2_ enhanced the antioxidant enzyme activities of *S. salsa* (SOD, CAT, and POD) under drought stress. In contrast, the antioxidant enzyme activities of *S. salsa* were significantly lower in the non-saline (CK) and MgCl_2_ treatments (Figure 8). These results indicated that Na^+^ and Ca^2+^ are essential for increasing the activity of antioxidant enzymes in *S. salsa*, thereby reducing drought stress. Despite the use of PEG6000 in this study to simulate drought stress, it was not able to completely mimic the natural drought conditions. Therefore, the physiological mechanism of salt and drought stress on the succulence and growth of *S. salsa* was further explored by manipulating the field moisture capacity in subsequent experiments.

## 5. Conclusions

These results combined provide evidence that Na^+^ and Ca^2+^ are the primary elements responsible for the succulence of *S. salsa* and can be used to combat drought stress. This is achieved by enlarging the size of leaf epidermal cells, resulting from an increase in leaf water content and a decrease in stomatal conductance, creating an imbalance between water absorption and drainage, as well as an increase in cell turgor pressure. Succulence can alleviate drought stress by maintaining the water content of *S. salsa* leaves. Simultaneously, Na^+^ and Ca^2+^ improve the activities of antioxidant enzymes (SOD, CAT, and POD) to alleviate oxidative stress. Therefore, succulence is a key factor in the response of *S. salsa* to salt and drought stress. Salt ions are essential for *S. salsa* growth and succulence.

## Figures and Tables

**Figure 1 plants-13-00721-f001:**
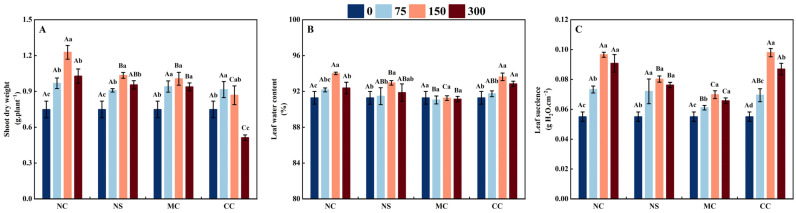
Shoot dry weight (**A**), leaf water content (**B**), and leaf succulence (**C**) of *S. salsa* as affected by various sodium and chloride salts. Note: NC, NaCl; NS, Na_2_SO_4_; MC, MgCl_2_; CC, CaCl_2_. Bars indicate ± standard error (SE) n = 3. Different lowercase letters above the bars indicate significant difference among salt concentrations under the same salt type, and different uppercase letters indicate significant difference among salt types under the same salt concentration (*p* < 0.05).

**Figure 2 plants-13-00721-f002:**
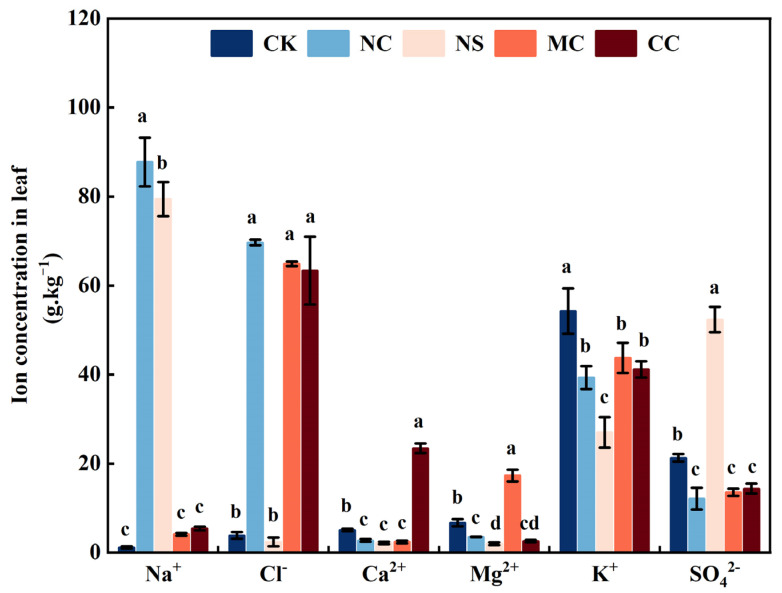
Different ion concentrations of *S. salsa* leaves affected by various sodium and chloride salts. Note: NC, NaCl; NS, Na_2_SO_4_; MC, MgCl_2_; CC, CaCl_2_. Bars indicate ± standard error (SE) n = 3. Different letters indicate a significant difference at *p* < 0.05.

**Figure 3 plants-13-00721-f003:**
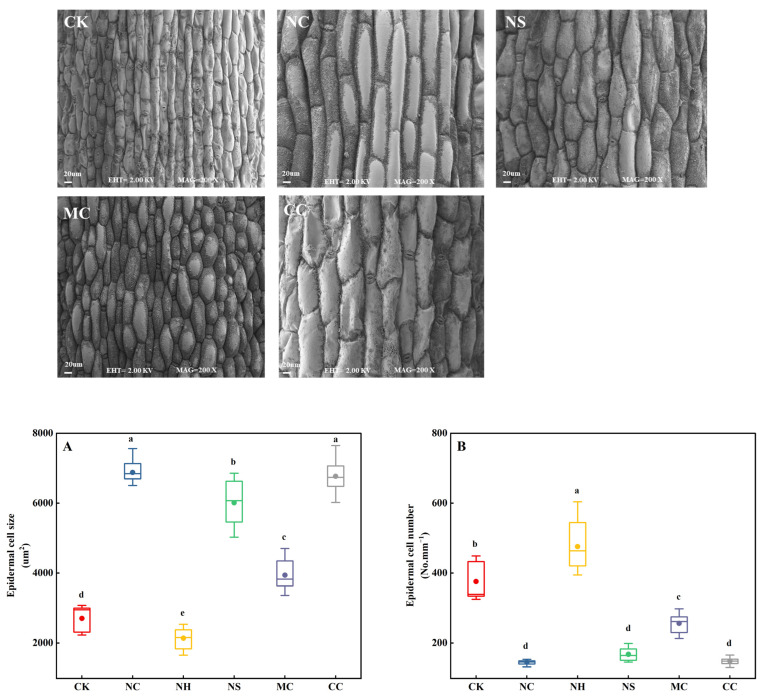
Epidermal cell size (**A**) and number (**B**) of *S. salsa* leaves affected by various sodium and chloride salts. Note: NC, NaCl; NS, Na_2_SO_4_; MC, MgCl_2_; CC, CaCl_2_. Bars indicate ± standard deviation (SD, n = 15). Different letters indicate a significant difference at *p* < 0.05.

**Figure 4 plants-13-00721-f004:**
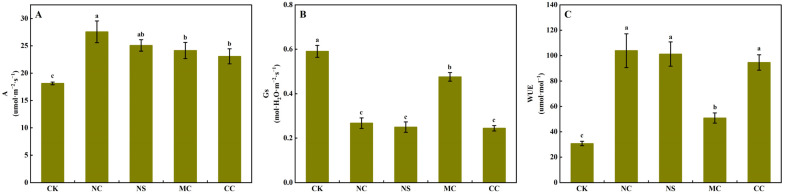
CO_2_ assimilation rate (**A**), stomatal conductance (**B**), and water use efficiency (**C**) of *S. salsa* leaves affected by various sodium and chloride salts. Note: A, CO_2_ assimilation rate; Gs, stomatal conductance; WUE, water use efficiency (A/Gs); NC, NaCl; NS, Na_2_SO_4_; MC, MgCl_2_; CC, CaCl_2_. Bars indicate ± standard deviation (SD, n = 6). Different letters indicate a significant difference at *p* < 0.05.

**Figure 5 plants-13-00721-f005:**
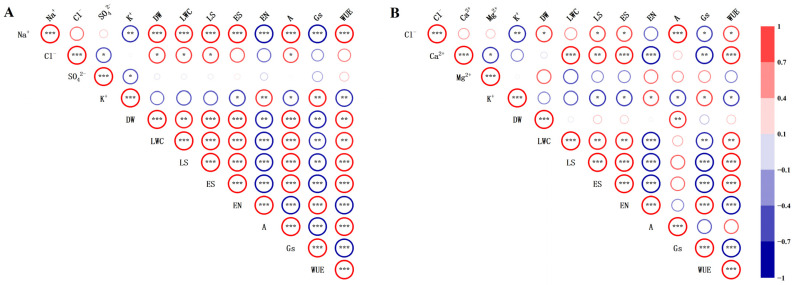
Correlation matrix for growth characteristics of *S. salsa* and various salt ions under different sodium salt (**A**) and chloride salt (**B**) treatments. Note: Gradient colour indicates the size of the Pearson correlation coefficient, ranging from −1 to 1. Blue indicates a negative correlation, and red denotes a positive correlation. Asterisks represent the level of significance and are shown only for *p* values: ** p* < 0.05, *** p* < 0.01, and **** p* < 0.001.

**Figure 6 plants-13-00721-f006:**
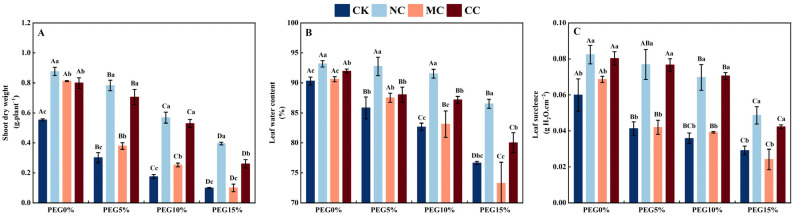
Shoot dry weight (**A**), leaf water content (**B**), and leaf succulence (**C**) of *S. salsa* affected by various salts and drought stress. Note: NC, NaCl; MC, MgCl_2_; CC, CaCl_2_. Bars indicate ± standard deviation (SD, n = 3). Different lowercase letters above the bars indicate significant difference among salt types under the same PEG concentration, and different uppercase letters indicate significant difference among PEG concentrations (*p* < 0.05).

**Figure 7 plants-13-00721-f007:**
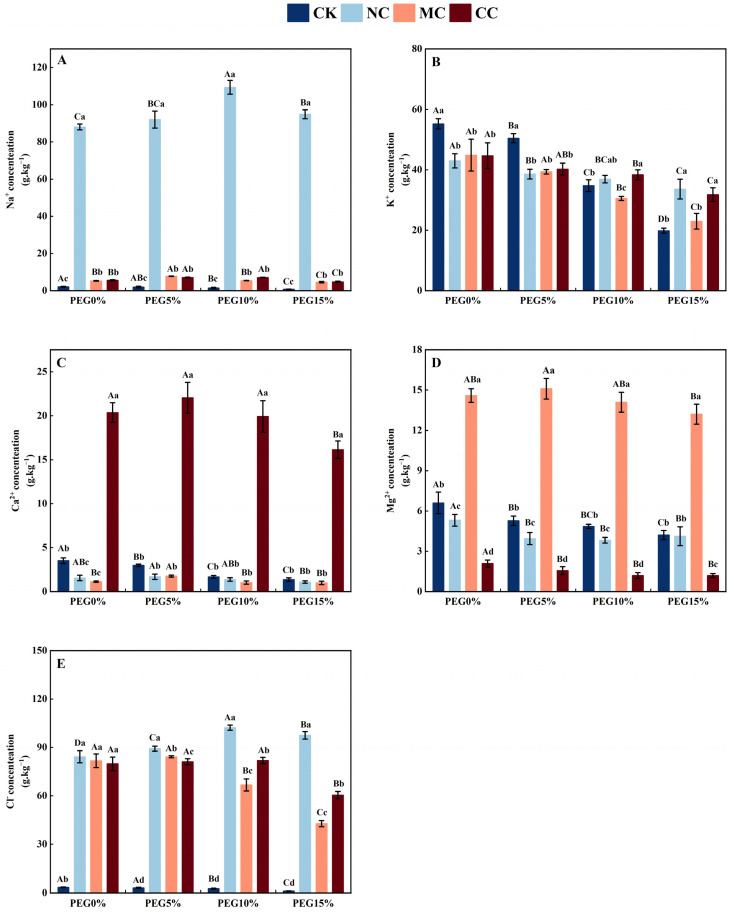
Sodium (**A**), potassium (**B**), calcium (**C**), magnesium (**D**), and chlorine (**E**) ion concentration of *S. salsa* as affected by various salts and osmotic stress. Note: NC, NaCl; MC, MgCl_2_; CC, CaCl_2_. Bars indicate ± standard deviation (SD, n = 3). Different lowercase letters above the bars indicate significant difference among salt types under the same PEG concentration, and different uppercase letters indicate significant difference among PEG concentrations (*p* < 0.05).

**Figure 8 plants-13-00721-f008:**
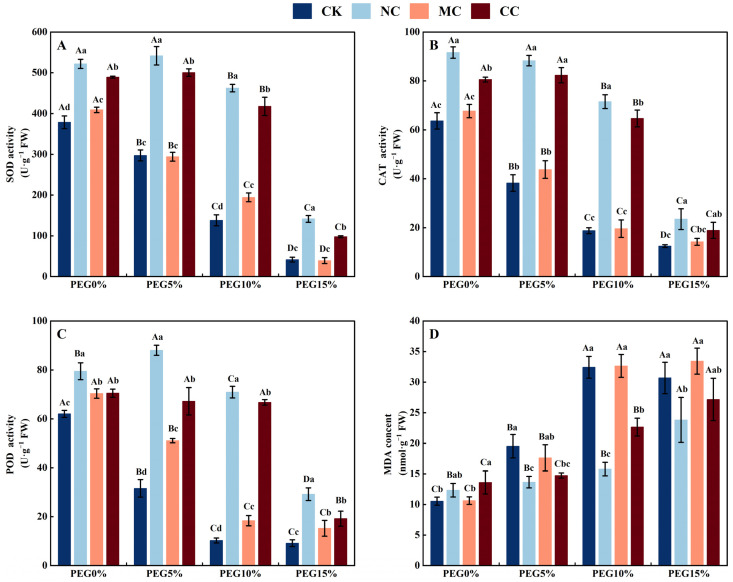
Antioxidant enzyme activities and malondialdehyde content of *S. salsa* as affected by various salts and osmotic stress. Note: NC, NaCl; MC, MgCl_2_; CC, CaCl_2_; SOD, superoxide dismutase (**A**); CAT, catalase (**B**); POD, Peroxidase (**C**); MDA, malondialdehyde (**D**). Bars indicate ± standard deviation (SD, n = 3). Different lowercase letters above the bars indicate significant difference among salt types under the same PEG concentration, and different uppercase letters indicate significant difference among PEG concentrations (*p* < 0.05).

**Figure 9 plants-13-00721-f009:**
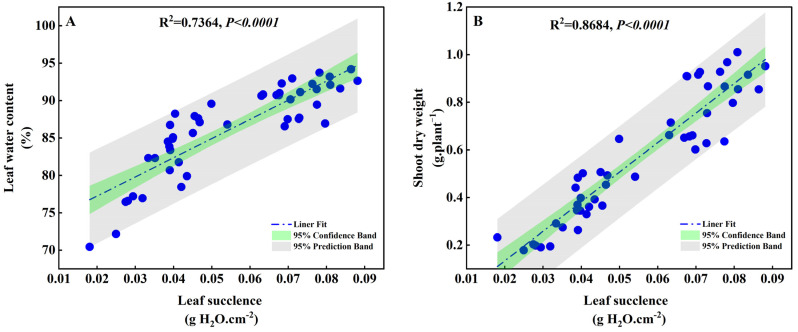
Correlation linear fitting for leaf succulence and leaf water content (**A**), and shoot dry weight (**B**) of *S. salsa*.

**Table 1 plants-13-00721-t001:** Two-way analysis of variance (ANOVA) testing the effects of PEG content (P) and salt ions (Si) on shoot dry weight, leaf succulence, leaf water content, ions concentration (Na^+^, K^+^, Ca^2+^, Mg^2+^, and Cl^−^), antioxidant enzyme activities, and malondialdehyde content of *S. salsa*.

Source	Two-Way ANOVA (F Value)											
	DW	LWC	LS	Na^+^	K^+^	Ca^2+^	Mg^2+^	Cl^−^	SOD	CAT	POD	MDA
PEG concentration (P)	737.45 ***	174.68 ***	123.85 ***	26.06 ***	150.05 ***	15.81 ***	15.69 ***	89.51 ***	1796.44 ***	779.96 ***	727.6 ***	158.67 ***
Salt ions (Si)	396.76 ***	73.44 ***	120.47 ***	9263.37 ***	11.73 ***	1379.25 ***	1168.46 ***	3353.91 ***	674.03 ***	380.03 ***	414.13 ***	29.1 ***
P × Si	21.42 ***	6.51 ***	4.78 **	24.37 ***	15.91 ***	5.38 **	2.21 *	64.82 ***	48.71 ***	38.62 ***	58.99 ***	9.85 ***

Note: DW, shoot dry weight; LWC, leaf water content; LS, leaf succulence; SOD, superoxide dismutase; CAT, catalase; POD, peroxidase; MDA, malondialdehyde. *p* < 0.05 *, *p* < 0.01 **, *p* < 0.001 ***; DF_P_ = 3, DF_St_ = 3, DF_P × St_ = 9.

## Data Availability

Data are contained within the article and Appendix A.

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
