# Peer review of "Exogenous Sodium and Calcium Alleviate Drought Stress by Promoting the Succulence of Suaeda salsa"

_plants, 2024, doi:10.3390/plants13050721_

Round 1

Reviewer 1 Report

Comments and Suggestions for Authors

the quality/size of figures should be improved as it is not readable. 

Measurements of epidermal cells of leaves: The leaf samples were collected and washed with distilled water to remove dust and salt. Subsequently, segments (5 mm long) were immersed in FAA solution, ... The leaves were then dehydrated using ascending concentrations of ethanol, critical-point dried:

what is FAA solution? why the  leaf dehydrated? the size of the cell is dependent on the water content? does it change and show variation? the method also need proper previous citation. 

Quantification of Na+ , K+ , Mg2+, Ca2+, Cl- and SO4 2- content? the method reported needs proper reference.

the method shows that the only the soluble component of these ions will be available, and water insoluble bound form of these ions will not be available? does it not effect the concentration values? 

Reviewer 2 Report

Comments and Suggestions for Authors

Although there are numerous publications on this species, the manuscript comprises numerous results obtained from a combination of treatments, not previously analyzed to my knowledge.  For this reason, the manuscript has merit but also a drawback. There are two different sets of experiments, apparently unrelated and therefore analyzed separately, which makes the manuscript look like a sum of two parts and without a main central idea. A working hypothesis is also missing.

The Material and methods section is unclear, for example the age of the seedlings in the two experiments. Also, since different analyses were performed in the two experiments, this should be reflected in this section.

In my opinion, this drawback can be solved by rearranging the Ms.

It is not clear whether the two experiments were performed simultaneously or whether the second experiment was performed after the first experiment.

In the case that the experiments were simultaneous, the same analysis should be performed on all variants of the experiment. As such, epidermal cell size and number and assimilation rate, stomatal conductance and water use efficiency should also be performed for seedlings in experiment 2.

If treatment 1 was in fact a screening for the selection of optimal salt concentrations, this should be indicated. Another aspect that is not explained at all is that shoot dry weight, leaf water content and succulence were analyzed with all tested concentrations, but all other analyses performed in experiment 1 included only one concentration.

The second experiment conducted with seedlings from a combination of salt and PEG treatments is interesting, but here also epidermal cell size and number, assimilation rate, stomatal conductance and water use efficiency were not analyzed. If possible, the addition of these data would improve the quality of the article. Otherwise, an explanation of the changes in the methodological approach in Experiment 2 is needed.

In summary, the manuscript needs careful reorganization and rewriting of the Material and Methods and Results sections. The inclusion of additional data on epidermal cell number size and assimilation rate, stomatal conductance, and water use efficiency in Experiment 2 would improve the quality of the manuscript.

Figure 1S should be included in the text, as the data presented therein are essential to the experimental design. Figure 3 (The PCA) should be deleted as it is not necessary for understanding the results.

The table should be inverted (change rows for columns).

For a better understanding, a flow chart of the experimental design would be useful.

Reviewer 3 Report

Comments and Suggestions for Authors

Manuscript Title: Exogenous sodium and calcium alleviate drought stress by pro- 3

moting the succulence of Suaeda salsa

The manuscript seems to be interesting and well presented with statistical data and figures it can be accepted once the minor issues are corrected.

1.      Abstract Keywords: Use catchy keywords which are not used in the title of the study.

2.      Choice of PEG in this study and their importance in drought evaluation in the plants can be shortly included in the introduction.

3.      Line 84-85: “However, the effects of various salt ions on euhalophyte succulence have received limited attention.” Kindly refer and cite the following reference about salt tolerance studies in Suaeda salsa-Ann Bot. 2015 Feb; 115(3): 541–553.10.1093/aob/mcu194.  And highlight the points in which way the present research is novel.

4.      Line 118: When was the seed collected? And how long the collected seeds were preserved under 4 degrees?

5.      Line 140: Kindly include the manufacture information and Analytical standard grade of the chemical polyethylene glycol (PEG-6000).

6.      Line 240: Legends of electron microscopy image was missing and mark and show the image to distinguish the size and quality of epidermal cells better understanding.

7.      Line 248: The binomial name in the legends should be in italics. Same in the figure legend four (Line 313).

8.      Usually PEG-induced stress does not completely replicate the natural conditions of drought stress that plants experience in their environment. The application of PEG primarily creates osmotic stress by reducing water availability, but it may lack some of the complexity and signaling pathways associated with actual drought stress. How can be defined this limitation and overcome? Kindly include key challenges and future perspectives of this present research in the conclusion.

Round 2

Reviewer 2 Report

Comments and Suggestions for Authors

In the new version of the MS, the experimental design is better explained. However, it remains unclear how many plants (replicates) were used for each experimental condition. Line 150 indicates that 19 leaves were collected, but it is necessary to specify how many plants were sampled. The same in line 176 "Each treatment group was replicated six times" . 

More information on enzyme and MDA quantification would also be of interest, not just naming the kit used.
